# Meta-Analysis and Systematic Literature Review of the Genus *Pneumocystis* in Pet, Farm, Zoo, and Wild Mammal Species

**DOI:** 10.3390/jof9111081

**Published:** 2023-11-04

**Authors:** Christiane Weissenbacher-Lang, Anna Grenl, Barbara Blasi

**Affiliations:** Department for Pathobiology, Institute of Pathology, University of Veterinary Medicine Vienna, Veterinärplatz 1, 1210 Vienna, Austria; anna.grenl@gmx.at (A.G.); barbara.blasi@vetmeduni.ac.at (B.B.)

**Keywords:** *Pneumocystis* species, animal, prevalence, detection method, level of infection, lung lesions, morphology, localization of fungus, immunosuppression, genetic information

## Abstract

A systematic literature search on *Pneumocystis* in 276 pet, farm, zoo, and wild mammal species resulted in 124 publications originating from 38 countries that were analyzed descriptively and statistically, for which inclusion and exclusion criteria were exactly defined. The range of recorded *Pneumocystis* prevalence was broad, yet in half of the citations a prevalence of ≤25% was documented. Prevalence was significantly dependent on the method used for *Pneumocystis* detection, with PCR revealing the highest percentages. Pet animals showed the lowest median *Pneumocystis* prevalence, followed by farm, wild, and zoo animals. In contrast, pet and farm animals showed higher proportions of high-grade infection levels compared to zoo and wild mammals. Only in individual cases, all of them associated with severe *Pneumocystis* pneumonia, was an underlying immunosuppression confirmed. Acquired immunosuppression caused by other diseases was frequently discussed, but its significance, especially in highly immunosuppressive cases, needs to be clarified. This meta-analysis supported a potential influence of the social and environmental factors of the host on *Pneumocystis* transmission in wildlife, which must be further elucidated, as well as the genetic diversity of the fungus.

## 1. Introduction

*Pneumocystis* (*P.*) spp. are a group of highly diversified, opportunistic fungi that are well adapted to the lungs of a wide variety of mammals [1]. *Pneumocystis* is an extracellular, highly host-specific pathogen that primarily attaches to alveolar epithelial cells. Proliferation is supported by immunosuppression and the fungus may fill pulmonary alveoli, a process which leads to respiratory failure [2]. Due to the large impact of *P. jirovecii* pneumonia, during the last decades, research has mainly concentrated on the different aspects of this disease in humans and experimental animals. Besides these species, the lungs of various animal species have also been investigated, and in these studies the focus was placed on the following topics: (a) *Pneumocystis* prevalence, (b) the comparison of the efficiency of detection methods, (c) the pathomorphological and histological descriptions of lung lesions, (d) the morphological description of the fungal organisms, and (e) the genetic characterization of single loci of different *Pneumocystis* species. In this review, we summarize the findings on *Pneumocystis* in 276 pet, farm, zoo, and wild animal species as published in 124 references. To draw conclusions about the published findings and to summarize them, a thorough meta-analysis was carried out. The main aim was the comparison of prevalent data from various mammal orders, but also within the categories of pet, farm, zoo, and wild animals. Furthermore, the impact of different detection methods on *Pneumocystis* prevalence was investigated. The level of infection was indicated as a criterion for distinguishing between subclinical infection and severe *Pneumocystis* pneumonia. The other aspects of *Pneumocystis* presence mentioned above were analyzed descriptively.

## 2. Materials and Methods

A systematic literature search according to the PRISMA guidelines [3] was conducted using the search engines Pubmed (https://pubmed.ncbi.nlm.nih.gov; accessed on 2 November 2023), Scopus (http://scopus.com; accessed on 2 November 2023), and Vetmedseeker (search-uvw.obvsg.at; accessed on 2 November 2023). The PRISMA checklist is provided in Appendix A. The inclusion criteria for the meta-analysis included (1) references in English, French, German, and Spanish, (2) the availability of the abstract and full text, (3) the investigation of pet, farm, zoo, or wild animal species, and (4) the availability of data on the main topics of this review. The main exclusion criteria were the investigation of humans, laboratory animals, or other animal species used in an animal trial. *Pneumocystis*, pneumocystosis, and names of zoological orders, families, and species in Latin and different languages were used as search terms. A total of 124 publications were analyzed and all data were checked by two persons. Since all studies that have ever been published on this topic were considered, there was no risk of bias assessment.

In the second step, the publications’ contents were reviewed and transferred into an Excel file. Each reference was considered as an individual citation (IC). In publications with several animal species, each animal species was recorded separately as an IC. Prevalence data from the same publication determined by different methods (e.g., PCR and serology in comparison) were considered separately. The information on the 539 ICs (with details in the data repository https://doi.org/10.34876/q34b-q773; accessed on 2 November 2023) was categorized as follows:Zoological order, suborder, family, and species;Housing category (pet, farm, zoo (also including primate centers, where no animal trial was carried out at that time), wild animal species);Breed;Age;Sex;Country of origin;*Pneumocystis* prevalence;Investigated sample number;Methods used for *Pneumocystis* determination;Level of infection (high-grade or low-grade infection level);Pathomorphological description of the lung lesions;Histopathological description of the lung lesions;Morphological description of *Pneumocystis* organisms (cytology, histology, electron microscopy) and their pulmonary and extrapulmonary localization;Clinical symptoms;Indication of immunosuppression or immunosuppressive concomitant factors;Genetic information.

For wild animals, additional data on social structure, habitat, and lifestyle were collected from the following sources: Tiere [4], animaldiversity.org (accessed on 2 November 2023), animalia.bio (accessed on 2 November 2023), eol.org (accessed on 2 November 2023), and gbif.org (accessed on 2 November 2023). The following parameters were included in the Excel file:Social structure (loner, pair, group);Group size;Habitat (forest, mountain, desert, grassland, marshland, river, mangrove swamp, coast, seaside, polar region, urban);Diet (herbivorous, granivorous, frugivorous, insectivorous, carnivorous, omnivorous, bloodsucking);Activity phase (diurnal, crepuscular, nocturnal, cathemeral);Lifestyle (terrestrial, semiaquatic, arboreal);Hibernation or torpor;Migratory lifestyle.

Besides a descriptive evaluation, statistical analyses were carried out using the software IBM SPSS Statistics version 27 (IBM Corporation, Armonk, NY, USA). The IC was the statistical unit. For the statistical analysis of *Pneumocystis* prevalence data, the following exclusion criteria were defined:Sample sizes ≤ 10 samples;Missing sample sizes or numbers of positive samples;Selected pre-defined samples or studies including only positive samples;Repeated studies using the same samples;Missing information on the exact species;Undefined methods and serological studies.

For statistical analysis of the level of infection, additionally, the following exclusion criteria were used:Studies with only negative results;Studies without the indication of the level of infection.

The final numbers of ICs used for statistical evaluation and exclusion criteria are summarized in Table 1, and a list including the respective references can be found in Appendix A. The parameters of the descriptive statistics’ mean value, median value, standard deviation, standard error, confidence interval, minimum, and maximum are provided in Appendix A. Data were first evaluated for their normal distribution using the Kolmogorov-Smirnov test, for variance homogeneity using Levene’s test, and for robustness using Welch’s test (Appendix A). To improve homogeneity and robustness, different methods to transform the normal distribution (square, square root, log, log10, reciprocal root, reciprocal, and reciprocal square of data values) were tested, but none of them improved the parameters significantly. Hence, non-parametric methods such as the Kruskal-Wallis and the χ^2^ test that are robust against a violation of the assumption of normality were used for further analy-ses. Differences in the mean *Pneumocystis* prevalence among the mammal orders, suborders, families, species, housing categories, investigated sample numbers, methods, and levels of infection were evaluated via the Kruskal-Wallis test (187 ICs). Social structure, habitat, and lifestyle parameters were additionally considered for wild animal species (133 ICs). Boxplots were chosen for the graphical presentation. Differences related to the level of infection among mammal orders, suborders, families, species, housing categories, investigated sample numbers, methods, and levels of infection were analyzed using the χ^2^ test (139 ICs). The social structure, habitat, and lifestyle parameters were additionally considered for wild animal species (58 ICs). The Kruskal-Wallis test was furthermore used for the analysis of associations between the prevalence and level of infection (60 ICs).

## 3. Results

### 3.1. Overview of Publications on Pet, Farm, Zoo, and Wild Animals Infected with Pneumocystis

The evaluated references cover 276 mammal species belonging to 51 mammal families from 13 mammal orders. The numbers of families, species, and ICs are summarized in Table 2. Two hundred and fifty-eight species were exactly defined, while in 18 cases the exact species name was not available. Table 3 contains an overview of all the mammal species analyzed in this work and the respective references.

#### 3.1.1. Examined Sample Size

The examined sample numbers varied strongly (Figure 1). In mammal orders Artiodactyla, Carnivora, Chiroptera, Perissodactyla, and Primates, *Pneumocystis* studies based on sample sizes below or equal to ten dominated. For representatives of orders Diprotodontia, Hyracoidea, and Pilosa, only individual case descriptions were available. In the Eulipotyphla, Lagomorpha, and Rodentia orders, at least half of the studies involved between 11 and 250 samples. In the cases of shrew tenrecs [5], greater grisons, Pallas’s long-tongued bat, murine mouse opossums, lowland pacas, Guyenne spiny-rats, and Guianan squirrels [74], the exact sample size was not indicated. A total of 62% of the species included in this study could be assigned to wild animals, 14% to zoo animals or animals living in primate centers, 14% to pet animals, and 9% to farm animals. In seven ICs, there was no information in this regard. For the determination of *Pneumocystis*, in all studies, except five that investigated bronchoalveolar lavage [20,36,49,55,100], lung tissue was used.

#### 3.1.2. Origins of Animals

The published mammal species originated from 38 different countries. Most ICs originated from European countries and, except for the order Didelphimorpha, all mammal orders were represented. In North America, a broad range of species was investigated, although a lower number of ICs was published. In South America, mainly members of the Chiroptera order, and in Africa and Asia, mainly members of the Rodentia order were investigated. Only a few ICs were published in Oceania (Figure 2).

### 3.2. Pneumocystis Prevalence

#### 3.2.1. *Pneumocystis* Prevalence in Studies with Sample Sizes > 10 Samples

In this meta-analysis, the prevalence data reported only reflect studies with sample numbers higher than ten (187 ICs). The overall prevalence ranged between one and ninety-three percent. In 43% of the ICs, a prevalence of ≤ 25% was recorded, while in 29% and in 5% of the ICs, 26–50% and over 75% of prevalence was documented, respectively. The Carnivora order showed the lowest median prevalence of 6% (based on *n* = 23 ICs), followed by Eulipotyphla (15%; *n* = 22), Perissodactyla (16%; *n* = 5), Lagomorpha (17%; *n* = 6), Artiodactyla (18%; *n* = 29), and Rodentia (27%; *n* = 76). The highest median prevalence was documented in the Chiroptera (30%; *n* = 23) and Primates (33%; *n* = 3) orders. A comparison of the arithmetic means revealed significant differences in prevalence (*p* = 0.043). The prevalence data of Carnivora, Eulipotyphla, and Rodentia showed higher standard deviations as the values were more widely dispersed (Figure 3).

Extraordinarily high prevalence data equal to or higher than 75% (mean sample number of 62, range of 26–203 samples) were achieved via PCR for *Pneumocystis* detection and were described for the:Carnivora order: Raccoon dog [70];Eulipotyphla order: Valais shrew [71];Rodentia order: European woodmouse [117], greater bandicoot rat [118,119], brown rat [118,119,123], and river rat [127].

Despite a mean sample number of 37 samples (range of 12–81 samples), *Pneumocystis* was not detected in several mammal species:Artiodactyla order: Cattle [9], domestic pig [9,21];Carnivora order: Dog [68], red fox, Eurasian badger [71], cat [73], and Siberian weasel [9];Chiroptera order: Rodriguez flying fox [82];Eulipotyphla order: Western European hedgehog [6], short-tailed shrew, Cinereus shrew, smoky shrew [85], and Eurasian pygmy shrew [93];Primates order: White-tufted ear marmoset [6];Rodentia order: Short-tailed field vole, common pine vole, bank vole, yellow-necked field mouse, European woodmouse [86], Smiths’s red-backed vole [115], California mouse, Western harvest mouse [87], Wilsons’s spiny mouse [114], and Eurasian field mouse [92].

A low mean prevalence below or equal to 5% (mean sample number of 68; range of 20–160 samples) was described in the:Artiodactyla order: Cattle, sheep, and domestic pig [11]Carnivora order: Dog, cat [21,64], and least weasel [75];Rodentia order: Short-tailed field vole, common vole, bank vole [10], California vole [87], brown rat [6], and river rat [71].

The prevalence was significantly dependent on the method used for *Pneumocystis* detection (Figure 4; p < 0.001), but not on the investigated sample size (*p* = 0.373). The highest median *Pneumocystis* prevalence of 50% was reached when using immunohistochemistry (IHC) (based on *n* = 3 ICs), followed by PCR (35%, *n* = 86), and histological examination (33%; *n* = 4). A lower prevalence was acquired via in situ hybridization (ISH) (15%; *n* = 16), and cytology (7%, *n* = 78).

The evaluation of prevalence data according to the four housing categories resulted in a low median *Pneumocystis* prevalence of only 9% in pet animals (based on *n* = 23 ICs), followed by 18% in farm animals (*n* = 27), 24% in wild animals (*n* = 133) and 33% in zoo animals (*n* = 3) (Figure 5; *p* = 0.069). The prevalence data of wild animals were furthermore analyzed regarding the influence of social structure and group size, habitat, diet, activity phase, a terrestrial, semiaquatic, or arboreal lifestyle, hibernation, and migratory behavior. Species living in woodlands (*p* = 0.046) and polar regions (*p* = 0.016) showed a significantly lower *Pneumocystis* prevalence. Omnivorous diets or herbivorous and frugivorous diets supplemented by insects or meat were associated with higher *Pneumocystis* prevalence (*p* = 0.037). The semiaquatic lifestyle was significantly related to higher *Pneumocystis* prevalence (*p* = 0.020). There was no significant association with other social parameters (*p* > 0.05).

#### 3.2.2. *Pneumocystis* Positive Rates in Studies with Sample Sizes ≤ 10 Samples

Studies with sample sizes of ≤ 10 samples were analyzed descriptively. Sixty percent (324/539) of the ICs belonged to studies where low sample numbers were used. They were mainly investigated in case reports, studies that covered different mammal species, or studies that focused on the comparison of different methods for *Pneumocystis* detection. If the sample number was low, it would likely happen that all samples tested were *Pneumocystis* positive (*n* = 99 ICs) or negative (*n* = 128). Prevalence data based on low sample sizes can be misleading and rarely reflect reality (*n* = 97).

Case reports with successful *Pneumocystis* detection were published for the:Artiodactyla order: Goat [12,13], sable antelope [14], llama [15], and domestic pig [33];Carnivora order: Dog [39,40,42,44,47,48,49,50,51,52,53,54,55,56,57,58,59,60,61,62,63,66,67], ferret [76], and American mink [77];Perissodactyla order: Horse [100,101,102,103,107,108,109,110];Pilosa order: Brown-throated sloth [111].

In several studies, low sample numbers of different mammal species were investigated. In one of the oldest studies of our meta-analysis, *Pneumocystis* was not detected in any of the seven investigated species using a Toxylin Hansen stain [7]. Cytological preparations stained with various special stains were furthermore used by Sebek & Rosicky (1967) [86] (0/5 species positive), Lainson & Shaw (1975) [78] (4/4 species positive), Poelma (1975) [37] (15/15 species positive), Yoshida & Ikai (1979) [9] (1/5 species positive), Settnes & Lodal (1980) [94] (1/7 species positive), Shimizu et al. (1985) [8] (3/5 species positive), Settnes et al. (1986) [16] (1/3 species positive), Shiota et al. (1986) [115] (1/3 species positive), Settnes & Henriksen (1989) [11] (0/4 species positive), Laakkonen (1998) [5] (4/4 species positive), and Laakkonen et al. (2001) [87] (0/2 species positive). Kucera et al. (1971) [10] summarized the results of their own studies in a review and described two positive mammal species via cytological investigation; in five other species, antibodies against *Pneumocystis* were determined. PCR was applied successfully in several recent studies on bats (Derouiche et al. (2009) [84] (1/1 positive species), Sanches et al. (2009) [81] (4/4 positive species), Akbar et al. (2012) [82] (8/14 positive species), Sanches et al. (2013) [79] (9/16 species positive), González-González et al. (2014) [83] (3/7 species positive), and Veloso et al. (2014) [80] (14/19 positive species)), monkeys (Demanche et al. (2001) [112] (16/16 species positive)), insectivores (Mazars et al. (1997) [88] (2/2 species positive)), and rodents (Danesi et al. (2016) [71] (1/2 species positive), Mazars et al. (1997) [88] (2/2 species positive), Latinne et al. (2018) [119] (5/5 species positive), Latinne et al. (2021) [118] (9/10 species positive), Petružela et al. (2019) [114] (7/27 species positive)). In a recent study, ISH revealed 23 *Pneumocystis* positive species out of a total of 84 investigated [6].

Two studies focused on the description and comparison of various methods for *Pneumocystis* detection in domestic pigs and foals [25,29].

### 3.3. Level of Infection

The level of infection is important to distinguish between *Pneumocystis* pneumonia and subclinical infection because it has already been shown that subclinically infected, asymptomatic individuals may serve as a reservoir [45] and that asymptomatic colonization likely represents the most typical manifestation in animal species [45,79]. The level of infection was reported in 26% (139/539) of the ICs. The infection level was either directly specified as “low-grade” or “high-grade” in the respective reference or morphological descriptions from histological or cytological examination (e.g., the presence of abundant amounts of honeycombed or foamy eosinophilic material) were used for classification into low- and high-grade infection levels in the present meta-analysis. In 41 ICs, only high-grade infection levels, and in 63 ICs, only low-grade infection levels were described. Thirty-five ICs contained cases with both low- and high-grade infection levels.

The distribution of low- and high-grade *Pneumocystis* infection levels among the mammal orders varied significantly (*p* < 0.001). Members of the Carnivora and Perissodactyla orders showed more high-grade than low-grade infection levels. Indeed, many case reports on severe *Pneumocystis* pneumonia in dogs [39,40,42,44,48,49,50,51,52,53,54,55,56,57,59,61,62,63,66,67] and horses [101,103,107,108,109,110] have been published. In three out of four ICs of *Pneumocystis* infection levels in the Pilosa order, high-grade infections were described [37,78,111]. In the Artiodactyla, Eulipotyphla, Lagomorpha, and Rodentia orders, low-grade infection levels dominated, and in Chiroptera only low-grade infection levels were documented. For orders Afrosoricida, Didelphimorphia, Hyracoidea, and Primates, no infection levels were published (Figure 6).

In pet animals, high-grade infection levels predominated, while in farm animals the proportion of low- and high-grade infection levels was comparable. In contrast, in zoo and wild animals, low-grade infection levels were mainly reported (Figure 7; *p* < 0.001). In pet animals, high-grade infection levels have been published in dogs [6,45,46], ferrets [6,76], rabbits, guinea pigs, and black rats [6], in addition to the case reports of dogs and horses previously mentioned. In farm animals, mainly domestic pigs and wild boars were associated with high-grade *Pneumocystis* infection levels [6,17,19,20,25,26,28,29,33,34,35,36]. Apart from domestic pigs, only two case reports of goats [12,13], and one of an American mink [77], have been published. Wild animals were represented by the Eurasian badger, Eurasian river otter, Oriental small-clawed otter [6], white-nosed coati [78], desert shrew and ornate shrew [87], European shrew [89,90,93], Laxmann’s shrew [93], European brown hare and mountain hare [96], brown-throated sloth [111], pale-throated three-toed sloth, Southern two-toed sloth, large-headed rice rat [78], and mouse [7].

Low-grade infection levels were associated with a median prevalence of 17%, and high-grade infection levels with one of 34% (Figure 8; *p* = 0.009). A statistical analysis of the correlation between the level of infection and clinical symptoms could not be conducted due to a lack of data regarding clinical symptoms.

For wild animals, parameters related to social structure, habitat, and lifestyle were evaluated. Compared to animals living in groups, loners showed a significantly higher number of cases with high-grade infection levels (*p* = 0.043). Group size, preferred habitat, diet, activity phase, a terrestrial, semiaquatic, or arboreal lifestyle, hibernation, and migratory behavior were not significantly associated with the level of infection (Figure 9; *p* > 0.05).

While mainly high-grade infection levels were described by means of histology, molecular-based methods with higher sensitivity, such as PCR and ISH, revealed large proportions of low-grade infection levels. Using IHC, the proportion of low- and high-grade infection levels was comparable. Cytology was mainly used in older studies when other techniques were not available, yet it proved to be a reliable tool for the detection of low-grade infections (Figure 10; *p* < 0.001).

### 3.4. Pathomorphological and Histopathological Description of the Lung Lesions

Five percent (26/539) of the ICs were presented together with a pathomorphological report. These ICs covered the mammalian species of goat [12], llama [15], domestic pig [18,25,26,29], dog [40,42,44,48,55,57,61,62,63], horse [25,98,99,101,103,106,107,109,110], and white-eared marmoset [37]. The consistency of the lungs was mainly described as firm or consolidated with no tendency to collapse [12,15,18,25,26,40,44,48,53,55,57,61,98,99,102,103,106,107,109,110]. In single cases, the lungs were associated with the attributes “rubbery” [26,44,48,62], “meaty” [25,99,101,103,107] or “hepatoid” [25,44,109,110]. The spectrum of colors ranged from red/purple [12,15,29,53,99,103,106,110] to white/yellow/brown [40,48,53,62,99,101,103] sometimes in a miliary-like or mottled pattern [42,53,61,62,63,99,101,103].

In seven ICs, no gross lesions were noticed [22,23,30,38,50,97,121]. Only in the study of Poelma & Broekhuizen (1972) [97] was the infection level described as low-grade. In all other studies, no infection level was indicated. In these cases, the fungal load was most likely low and therefore did not cause macroscopically detectable lung lesions. In three studies, different pneumonia types were reported [22,30,37].

The absence or presence of histological lung lesions was mentioned in 111 ICs. Lung lesions consistent with an interstitial pneumonia were described in 77 ICs on *Pneumocystis* in various mammal species. If stated, the affected animals showed different levels of infection [6,17,25,28,34,35,40,42,44,48,52,53,54,55,56,57,61,62,63,65,66,67,76,77,78,98,99,101,102,103,107,108,109,110,111]. In one case of a sable antelope, a moderate focal pneumonia with a low-grade *Pneumocystis* infection level was mentioned [14]. Mild histological lesions without further detailed description, all resulting from low fungal loads, were documented in five ICs on *Pneumocystis* in different insectivore and rodent species [86,115]. A granulomatous or purulent component was described in single cases of domestic pigs [6,19,34,35], dogs [6,53], Oriental small-clawed otters [6], cats [73], guinea pigs, and black rats [6], including some with high grade-infection levels. In one IC, the necrosis of alveolar and bronchial epithelium was documented in the lungs of a horse. In this case, the level of infection was not indicated [106]. In 26 ICs, no histological lesions consistent with *Pneumocystis* pneumonia were documented. Only the case of one dog was associated with a high-grade infection level [50]. In the other ICs, the infection level was either low [6,64,75,86,92], not indicated [21,124,127], or the animals were tested negative for *Pneumocystis* [21,86].

### 3.5. Morphological Description of *Pneumocystis* Organisms and Their Pulmonary and Extrapulmonary Localization

The general cytologic description of *Pneumocystis* asci refers to multinucleate bodies, approximately the size of red blood cells, containing eight nuclei. Trophozoites are described as slightly eosinophilic, empty globular structures. Histologically, *Pneumocystis* occurs as a foamy eosinophilic material or eosinophilic spherical bodies free in the alveolar spaces. This intracellular presence within alveolar macrophages was described in domestic pigs [18], dogs [53,67], horses [99,100,102,108,110], and a ferret [76]. Extrapulmonary pneumocystosis is only observed sporadically. The fungus spreads via both lymphatic and hematogenous routes and mainly affects the lymph nodes, spleen, liver, and bone marrow [128]. In non-laboratory animals, single cases of extrapulmonary pneumocystosis have been described only in dogs. The oldest study on *Pneumocystis* describes the involvement of the lungs, heart, and lymph node of a 9-week-old German shepherd [63]. Weissenbacher-Lang et al. (2017) [67] published the case of a 3.5-year-old Whippet mixed-breed suffering from demodicosis with a lymphogenous spread of the fungus. In a 1.5-year-old toy poodle with an assumed immunosuppression, *Pneumocystis* organisms were detected in the lungs, lymph nodes, liver, heart, kidneys, spleen, gastrointestinal tract, and pancreas [62]. The investigation of cytological preparations of a 16-month-old Chihuahua confirmed the presence of the fungus in the lungs, hepatic lymph node, liver, and spleen [53].

Cytological preparations were stained with:Giemsa [8,12,21,23,37,39,51,69,78,85,86,97,115];Toluidine blue O [8,9,11,16,23,50,64,88,94,108,109,112];Grocott’s methenamine-silver nitrate (GMS) staining [12,33,37,39,49,53,65,73,75,85,87,90,91,92,93,97,99,102,106,108,109];Periodic acid–Schiff (PAS) [37,85,99,106,108];Toxylin Hansen [7];Rapid Romanowsky Stain [49];Hema-Diff [59];Diff-Quik staining [45].

Only the internal structures of the cysts stain weakly with H&E; cyst walls and trophozoites remain unstained [129]. In addition to the H&E staining, histological preparations were stained with:Giemsa [12,13,48,52,57,63,104];GMS [6,12,15,18,19,25,29,40,42,44,48,52,54,56,61,62,67,77,89,98,101,103,104,107,110,111,125];PAS [26,29,40,44,54,56,61,104];TBO [104].

Chromogenic and fluorescent ISH stain all developmental stages of Pneumocystis. These methods were established either on the 18S ribosomal RNA (rRNA) [19,25] or on the 5S rRNA [29] gene and were successfully used for the detection of different *Pneumocystis* species:Artiodactyla order: Blackbuck, bison, water buffalo, cattle, goat, sheep, chamois, Bactrian camel, alpaca, Western roe deer, deer, wild boar, and domestic pig [6,19,25,29,35];Carnivora order: Dog, gray wolf, Eastern Canadian wolf, cat, Oriental small-clawed otter, Northern American river otter, European mink, beach marten, Eurasian badger, ferret, Eurasian river otter, striped skunk, and raccoon [6,67,68];Chiroptera order: Particolored bat [6];Eulipotyphla order: European shrew [6];Lagomorpha order: European brown hare and rabbit [6];Perissodactyla order: Horse [6,25];Rodentia order: Guinea pig, long-tailed chinchilla, black-bellied hamster, brown rat, and black rat [6].

ISH showed that *Pneumocystis* organisms were primarily located in the alveoli. In lungs with low ISH scores, only a few scattered *Pneumocystis* spp. organisms were attached to the alveolar wall, whereas a continuous lining of alveolar spaces by the organisms and a filling of the alveoli with the fungus was visible in moderately to severely infected lungs. The distribution pattern of *Pneumocystis* spp. in severely infected domestic pig lungs varied. In some cases, large clusters of the fungal spheroids were focally observed, but diffuse distribution patterns also occurred [6].

Comparable descriptions were achieved using IHC. The cross-binding of different commercially available anti-*Pneumocystis jirovecii* or anti-*Pneumocystis carinii* antibodies was used to label *Pneumocystis* organisms and describe their morphological distribution in the lungs of domestic pigs [25,27], wild boars [17], dogs [61,62], and horses [25,103]. Only Laakkonen & Sukura (1997) [90] used this technique to quantify *Pneumocystis* organisms in lung homogenates from common shrews.

Transmission electron microscopy was applied in 15 ICs of the mammal species goat [12], llama [15], domestic pig [29,33], wild boar [17], dog [40,42,48,56,61], least weasel [75], common shrew [90], and horse [108,109,110]. This technique was used for the description of *Pneumocystis* developmental stages, the localization of the fungus, and of cell lesions. The size of trophic forms was indicated as 1–5 µm for goat-derived [12], 3–5 µm for domestic pig-derived [29], 2–5 µm for wild boar-derived [17], and 1–4.5 µm for horse-derived *Pneumocystis* species [108]. The pleomorphic to round-shaped trophozoites were surrounded by a thin wall consisting of two membranes and contained endoplasmic reticular membranes, numerous glycogen particles, and lipid-like bodies [12,40,48,61,108,109,110]. The pellicle was folded to filopodia and the type 1 pneumocytes had cytoplasmatic projections that partially surrounded the trophozoites [29,33,42,48,110]. McConnell et al. (1971) [12] described the presence of adjoining trophozoites that communicated through a small opening in the pellicle. The trophic stages usually outnumbered the asci [12,29,48,56,90]. Cystic forms of *Pneumocystis* species isolated from domestic pig [29] and wild boar [17] measured 3–5 µm in diameter, those of common shrews, Laxmann’s shrews, and short-tailed field voles measured 3.5–3.9 µm [90], and those of horses measured 1.5–5 µm [108]. They were round to ovoid, surrounded by a 50–200 nm thick cell wall, and contained up to eight large intracystic bodies [12,15,17,29,40,56,61,108,109,110]. The occurrence of crescent-shaped asci was also reported, and the authors assumed that these had ruptured, expelling the intracystic bodies, since trophozoites were found in the direct vicinity and still partly surrounded by the asci [12,40]. The alveolar surfaces were covered by single or multiple layers of *Pneumocystis* organisms [15,17,29,40,42,48,108,109] that were also localized on the bronchial epithelium [33,40] and inside macrophages [42,48,108]. The ultrastructural cell lesions varied and both preservation of the morphologic integrity of the epithelium [48,108] and cell degeneration [108] or inflammatory reaction [40] were observed.

### 3.6. Indication of Immunosuppression or Immunosuppressive Concomitant Factors

In *Pneumocystis* positive dogs and horses, immunosuppression was proven either by determining serum protein levels and verifying hypogammaglobulinemia [52,54,55,57,60,101,102,104] or by proving the absence of B cells in the lymphoid tissue using IHC [54,55,58,62,101,102]. Breeds such as Cavalier King Charles spaniels [47,49,52,57,59,60,61], miniature dachshunds [40,44,48,55,56], or Arabian horses [104,109] have been associated with congenital immunosuppression and may, therefore, be susceptible to *Pneumocystis* pneumonia already at an early age. A potential association between *Pneumocystis* pneumonia and co-infections, some involving highly immunosuppressive agents, was contemplated in 120 ICs in the:Artiodactyla order: Cattle, goat, sheep, blackbuck, bison, chamois, water buffalo, alpaca [6], sable antelope [14], llama [15], Western roe deer [6,16], domestic pig [6,19,20,24,26,27,28,29,30,33,35,36], and wild boar [6,17];Carnivora order: Lesser panda [37], dog [6,46,50,51,54,60,65,67], Fennec fox [37], cat [6,72], gray wolf, Eastern Canadian wolf, beach marten, Eurasian badger, ferret, Eurasian river otter, Oriental small-clawed otter, Northern American river otter, striped skunk, European mink, raccoon [6], least weasel [75], and American mink [77];Chiroptera order: Wagner’s bonneted bat, Pallas’s mastiff bat, black mastiff bat, broad-eared bat, big free-tailed bat, brown mastiff bat, fringed fruit-eating bat, common vampire bat, Western red bat, Yellowish myotis, black myotis [80], Mexican great funnel-eared bat, hairy fruit-eating bat, California myotis [83], Brazilian free-tailed bat, and Pallas’s long-tongued bat [80,83];Diprotodontia order: Red kangaroo [37];Eulipotyphla order: European shrew [6,91,92], Laxmann’s shrew [92], and common tree shrew [37];Hyracoidea order: Southern tree hyrax and cape-rock hyrax [37];Lagomorpha order: European brown hare [6,16,96], mountain hare [96], and rabbit [6];Perissodactyla order: Horse [6,98,104,106,107,109];Pilosa order: Brown-throated sloth [111];Primates order: Brown howler monkey, Senegal-Galago, Demidoff’s Galago [37], Goeldi’s marmoset, Geoffroy’s marmoset, white-tufted ear marmoset, brown-headed tamarin, emperor tamarin, Midas tamarin, cotton-top tamarin, common squirrel monkey, Allen’s swamp monkey, owl-faced monkey, white-nosed guenon, crab-eating macaque, rhesus monkey, pig-tailed monkey, white-faced saki, and bamboo lemur [112];Rodentia order: Kangaroo rat, deer mouse [116], guinea pig, long-tailed chinchilla, black-bellied hamster, brown rat, black rat [6], yellow-necked field mouse, European harvest mouse [92], mouse [121], brown rat, and black rat [125,126].

Stress [100], the lack of uptake of colostrum [110], and the application of corticosteroids [101] were indicated as non-infectious causes for immunosuppression in horses. Others were different forms of neoplasia, reported in the *Pneumocystis* positive Bactrian camel, dog, and ferret, diabetes mellitus, which is also discussed as contributing to immunodeficiency, in a dog and a ferret [6], and tetralogy of fallot with polycythemia in a ferret [76].

In only 34% (48/140) of the ICs indicating the status of immunosuppression or potentially immunosuppressive concomitant factors, high-grade infection levels were confirmed by different diagnostic tools. In all ICs in which the immune status was investigated, except for two where the infection level was not indicated [58,102], high-grade infection levels were documented, and *Pneumocystis* pneumonia was associated with immunodeficiency. Immunosuppressive pathogens, such as porcine reproductive and respiratory syndrome virus (PRRSV), porcine circovirus 2 (PCV2), classical swine fever virus, canine distemper virus, canine parvovirus, feline panleukopenia virus, feline leukemia virus, mycoplasmas, *Demodex canis*, or *Histoplasma* spp., were reported.

### 3.7. Genetic Information

Currently, the genome landscape for most of the *Pneumocystis* species is almost completely unknown. The genomes of *Pneumocystis* species derived from humans and laboratory animals have been studied thoroughly [43,130]. In pet animals, only the genome of *P. canis* (dog), has been published [43]. The whole or partial genomes of *P. carinii* (rat), *P. wakefieldiae* (rat), *P. murina* (mouse), *P. oryctolagy* (rabbit), and *P*. sp. *macacae* (macaque) were described based on samples of laboratory animals [43,130]. Partial sequences of smaller genome segments proved that *Pneumocystis* species isolated from the corresponding domestic or wild animals do not differ genetically. However, no whole *Pneumocystis* genome is yet available from pet or wild animal isolates. Regarding *Pneumocystis* species derived from pet, farm, zoo, and wild animals, mainly the mitochondrial large and small subunit rRNA (mLSU and mtSSU rRNA) genes have been sequenced or used in phylogenetical characterization in the:Artiodactyla order: Cattle, sheep, chamois [6], goat [6,13], pig [6,22,29,32], and wild boar [6];Carnivora order: Dog [6,45,46,47,53,54,60,62,67], red fox [70,71], raccoon dog [70], golden jackal [38], cat [72], and greater grison [74];Chiroptera order: Brazilian free-tailed bat, common pipistrelle [82,84], Parnell’s mustached bat, Egyptian rousette, Rodriguez flying fox, common serotine, California myotis, noctule, brown big-eared bat, gray big-eared bat [82], Pallas’s long-tongued bat [74,82,84], and particolored bat [6];Didelphimorphia order: Murine mouse opossum [74];Eulipotyphla order: Valais shrew [71], European shrew [6,88], English and Finnish shrew species [95], and European mole [88];Lagomorpha order: European brown hare and rabbit [6];Perissodactyla order: Horse [6,105];Primates order: Goeldi’s marmoset, Geoffroy’s marmoset, white-tufted ear marmoset, brown-headed tamarin, emperor tamarin, Midas tamarin, cotton-top tamarin, common squirrel monkey, Allen’s swamp monkey, owl-faced monkey, white-nosed guenon, rhesus monkey, pig-tailed macaque, bamboo lemur, white-faced saki [112], and crab-eating macaque [112,113];Rodentia order: Short-tailed field vole [88], bank vole, yellow-necked field mouse [71], lowland paca, Guyenne spiny-rat [74], European woodmouse [88,117], mouse [122], Mount Banahao forest mouse, Neill’s Leopoldamys, long-tailed giant rat, shrew mouse, Malayan field rat, Mueller’s giant Sunda rat, hoary bamboo rat [118], greater bandicoot rat, Savile’s bandicoot rat, Berdmore’s Berylmys, Bower’s white-toothed rat, long-tailed giant rat, Indomalayan maxomys, Ryukyu mouse, fawn-colored mouse, Cook’s mouse, chestnut white-bellied rat, Indochinese forest rat, rice-field rat, Polynesian rat, white-footed Indochinese rat, lesser rice field rat, Oriental house rat, lesser bamboo rat [118,119], brown rat [6,118,119,123,124], river rat [71,127], garden dormouse [88], Finlayson’s squirrel [127], and Guianan squirrel [74].

Sequences of the DHPS locus are available for primates and rodents:Primates order: Goeldi’s marmoset, Geoffroy’s marmoset, white-tufted ear marmoset, brown-headed tamarin, emperor tamarin, Midas tamarin, cotton-top tamarin, common squirrel monkey, Allen’s swamp monkey, owl-faced monkey, white-nosed guenon, rhesus monkey, pig-tailed macaque, bamboo lemur, white-faced saki [112], and crab-eating macaque [112,113];Rodentia order: Mount Banahao forest mouse, Neill’s Leopoldamys, long-tailed giant rat, shrew mouse, Malayan field rat, Mueller’s giant Sunda rat, hoary bamboo rat, greater bandicoot rat, Savile’s bandicoot rat, Berdmore’s Berylmys, Bower’s white-toothed rat, long-tailed giant rat, Indomalayan maxomys, Ryukyu mouse, fawn-colored mouse, Cook’s mouse, chestnut white-bellied rat, Indochinese forest rat, rice-field rat, Polynesian rat, white-footed Indochinese rat, lesser rice field rat, Oriental house rat, lesser bamboo rat, brown rat [118], silvery mole-rat, woodland dormouse, dormouse, Nguru spiny mouse, Muze spiny mouse, fiery spiny mouse, Wilson’s spiny mouse, Kilonzo’s brush furred rat, Makundi’s brush furred rat, East African gerbil, Hinde’s rock rat, red rock rat, Kaiser’s rock rat, African woodland thicket rat, typical striped grass mouse, single-striped grass mouse, Mesic four-striped grass rat, Southern African pygmy mouse, gray-bellied mouse, Angoni vlei rat, Arc Mountain wood mouse, African soft-furred rat, Brockman’s Myomyscus, delectable soft-furred mouse, pouched mouse, fat mouse, and East African mole rat [114].

One study investigated the lung transcriptome of pigs [24] and another one the mycobiome of the lungs of kangaroo rats and deer mice [116].

## 4. Discussion

The range of published *Pneumocystis* prevalence data was very broad (1–93%). Since this was not a classical meta-analysis comparing two clinical groups, the application of the meta-analysis function in IBM SPSS could not be used; therefore, no statistical evaluation on the bias of missing results could be carried out. The analysis of data homogeneity and robustness was still carried out and non-parametric tests that are robust against a violation of the assumption of normality were used. Prevalence data below or equal to 25% predominated. In some mammal orders, prevalence data showed a wide dispersion that was statistically not related to the sample size. Accordingly, the method used for *Pneumocystis* detection had an influence, especially PCR, as a highly sensitive method revealed the highest prevalence data. The second most used molecular-based method was ISH, but this yielded significantly lower prevalence data. This could be due to the lack of amplification in the methodology compared to the PCR. Among the non-molecular-based methods, high prevalence data could be achieved by means of histology. In contrast, the median *Pneumocystis* prevalence obtained by cytology was lower but showed a wider dispersion compared to histology. Extraordinarily high prevalence data were achieved via PCR in wild carnivores, insectivores, and rodents, while negative results or low prevalence data were obtained via cytology. Negative prevalence results were obtained through different methods and could be related to an inadequate sample size, the level of infection, or the sensitivity of the method. In cases of molecular methods, the inability to design fully specific primers and probes due to a lack of information on the *Pneumocystis* genomes of most mammal species may be a pitfall. Techniques such as IHC, which is based on antigen detection using specific antibodies, may be problematic since only anti-Pneumocystis antibodies targeting human or rat *Pneumocystis* are commercially available and cross-binding with other *Pneumocystis* species has only been proven for domestic pigs [25,27], wild boars [17], dogs [61,62], horses [25,103] and the common shrew [90]. Nevertheless, *Pneumocystis* species differ genetically and antigenically [1] which may have a negative impact on the reliability of diagnostic methods. Regarding social and environmental factors in wild animals, the habitat may have an influence since mammal species living in woodland or polar regions were associated with a lower *Pneumocystis* prevalence. This would correspond to the published data of wild Asian rodents that showed a higher *Pneumocystis* prevalence in areas with habitat fragmentation and landscape patchiness. The authors explained this with a higher rodent diversity and density [118]. Poelma & Broekhuizen (1972) [97] found no association with the habitats of European brown hares. Akbar et al. (2012) [82] documented a 33-fold higher probability of picking a Pneumocystis-infected bat in a crowding species compared to non-crowding ones. A negative correlation between *Pneumocystis* prevalence and host density was described by Laakkonen (1995) [93] in European shrews. Due to a lack of data, the influence of climate or season on *Pneumocystis* prevalence could not be evaluated in the present meta-analysis. Only few studies addressed this topic. Laakkonen (1995) [93] described a higher *Pneumocystis* prevalence in European shrews in autumn, whereas spring was associated with the highest prevalence in the European shrew and the Alpine shrew [86], and in the large Japanese field mouse [115]. Elevation and precipitation favored *Pneumocystis* infection in various Asian rodent species [118] and crab-eating macaques [131], whereas different temperatures or relative humidity had no influence [131]. In bats, no impact of climate and elevation on *Pneumocystis* prevalence was found [82]. The present meta-analysis revealed a positive effect of omnivorous or herbivorous and frugivorous diets supplemented by insects or meat on *Pneumocystis* prevalence. In bats, no influence of food regimen was determined [82]. These authors also analyzed the impact of roosting habits and mating systems but found no association.

More than half of the studies used sample numbers below or equal to ten. The aim of these studies was mainly the description of individual *Pneumocystis* pneumonia cases, successful *Pneumocystis* detection in different mammal species, the morphological description of the fungus and related histological lesions, or the comparison of different detection methods.

Despite its great significance, the level of infection was rarely indicated. In the Carnivora and Perissodactyla orders, high-grade infection levels dominated. For these orders, mainly single *Pneumocystis* pneumonia cases with severe clinical symptoms and high fungal loads have been published. Only in dogs [52,54,55,57,58,60,62] and horses [101,102,104] has immunosuppression been documented. The evaluation of ICs according to the groups of pet, farm, zoo, and wild animals resulted in a higher proportion of high-grade infection levels, especially in pet animals, and followed by farm animals. In pet animals, new breeds are continuously established, and in farm animals, established breeds are selectively bred for better performance. Genetic defects primarily result from selective breeding for certain traits. Very little is known about congenital immunosuppression in animals and their underlying causes, but an association between selective breeding and congenital immunosuppression cannot be ruled out. Nevertheless, high-grade infection levels with *Pneumocystis* were also reported in wild animals. Impairment of the immune system in this case could be the result of a limited habitat, a reduced or suboptimal food supply, or inbreeding. Wild animals living in groups showed less high-grade infection levels, which may be a consequence of group/herd immunity. Especially in loners, high *Pneumocystis* amounts may also be required to ensure transmission of the fungus to the next host. In contrast, high-grade infection levels were significantly associated with higher prevalence. This could be due to a stronger transmission of the fungus from heavily infected lungs to other animals. The influence of social and environmental factors on *Pneumocystis* transmission has been only superficially studied. There are no studies considering different levels of infection. Especially in wild animals, potential correlations are difficult to prove. Last, but not least, the determined infection level depends directly on the sensitivity and the ability of quantification or semi-quantification of the method used for *Pneumocystis* detection.

In most of the studies, either no clinical symptoms or no detailed description of symptoms were reported, which could be mainly explained by the study design. Hence, this parameter was not considered in the statistical analysis. *Pneumocystis* pneumonia is typically associated with interstitial pneumonia. The severity of lung lesions may depend on the fungal load [132] and the present meta-analysis showed that in cases with low *Pneumocystis* amounts, the lungs may even not be impaired at all by the infection. Whole genome sequencing of *Pneumocystis* species derived from laboratory animals confirmed a substantial reduction in many metabolic pathways and a dependence of the fungus on oxidative phosphorylation, suggesting that energy production largely relies on glucose utilization through oxidative pathways [130]. Since many characteristics of *Pneumocystis* species suggested a biotrophic existence within the lungs of the mammalian hosts [133], a compromised lung environment may be detrimental to the survival of the fungus.

In both cytology and histopathology, special stains are mandatory to confirm the presence of *Pneumocystis* organisms [134]. In cytologic smears as well as in paraffin-embedded tissue samples, standard Giemsa, Wright, or H&E stains should be supplemented by GMS stain, which is known to be more sensitive [134]. GMS-stained slides are probably easier to evaluate than slides stained with any other method, because of the higher color contrast of the black cyst walls against the mainly green background. However, the disadvantage is that only cysts are stained positively, whereas the larger proportion of *Pneumocystis* organisms, the trophozoites, remain unstained, which could be a problem in cases with a low number of organisms present [129]. Also, with PAS stain, only the cell walls of cystic stages are captured. Additionally, the evaluation is more complex because cell walls and surrounding tissue are both stained in different shades of pink and red [129]. Polychrome stains, such as Giemsa and Diff-Quik [135], stain the nuclei pink-purple and the cytoplasm blue and are, for this reason, easier to evaluate. *Pneumocystis* cell walls are not stained and appear as a clear halo around the cystic forms. The biggest advantage of these methods lies in staining trophic forms, which remain unstained in GMS and PAS stains [135]. Compared to GMS, Diff-Quik has a significantly lower sensitivity, but is a relatively simple procedure that gives quick results. The application of an appropriate special staining method enhances a correct cytologic or histologic diagnosis. However, the evaluation of cytological preparations requires experience, especially when the numbers of *Pneumocystis* organisms are low because the typical structures can be missed easily. In humans, the pathogen is not visualized in 32–80% of *Pneumocystis* pneumonia cases [136] and in dogs, *Pneumocystis* has not been detected reliably in BALF [57,61,137].

Methods such as ISH and IHC allow the correlation of the localization of the fungus with the histological lung lesions [6,19] and can be used for the confirmation of results acquired in vitro [34]. Compared to special stains, especially of cytological preparations, the substantially higher technical effort and costs are a disadvantage, though.

Transmission electron microscopy is beneficial for the description of the fungus, and also of host cell lesions. However, the technique is complex and requires detailed planning and coordination of sampling and sample preparation. This is not always possible when taking samples from species that are not laboratory animals.

Around the time of World War II, a new lethal pneumonia in infants between two and eight months of age occurred. Premature or malnourished infants from orphanages were mainly affected and cases of the so-called interstitial plasma cell pneumonitis were described in many European countries [138,139,140,141]. This disease was not associated with *Pneumocystis carinii* until 1942, when van der Meer and Brug detected cysts in the alveoli of three Dutch infants with pneumonia [7]. For the next decade, *Pneumocystis* pneumonia was therefore considered limited to infants. In the 1950s, the population of immunocompromised patients was exceptionally low because immunosuppressive therapy and the use of cortisone were in their initial stages and organ transplantation was not yet an established medical practice. The first indications of a link between immunosuppression and *Pneumocystis* pneumonia were found in inoculated and uninoculated rats and rabbits, which developed severe pneumonia after treatment with cortisone [142,143]. Pentamidine isoethionate was the first effective drug used against *Pneumocystis* pneumonia [144] and remained the standard therapy until the 1970s. The drug was only provided by the Parasitic Disease Drug Service at the Centers for Disease Control after the indication of clinical data of the patients by the respective physician. Thus, the first *Pneumocystis* database emerged, and cancer was identified as an underlying disease for this type of pneumonia [145]. With an increase in the use of immunosuppressive anticancer treatment, the prevalence of *Pneumocystis* pneumonia also increased [146]. During the following years, the disease was also associated with primary immunodeficiency disorders [147,148] and the ablation of the immune system for organ transplantations [149]. In the 1980s, the list of risk factors was extended by the newly discovered human immunodeficiency virus (HIV) [150] and within a few months it became obvious that *Pneumocystis* was one of the major infecting agents in AIDS patients [151,152,153].

Risk factors for clinical *Pneumocystis* pneumonia in animal species have only rudimentarily been discussed. Immunosuppression in animals was only proven in a few ICs of dogs and horses and was mainly associated with high-grade infection levels. Little is known about congenital immunosuppression in animals and severe *Pneumocystis* infections have only been associated with some dog and horse breeds. These cases were also mainly related to high-grade infection levels. Several studies described the detection of *Pneumocystis* and various other pathogens and discussed the acquired immunosuppression caused by them. If the immune status of the animal is not evaluated, a definite conclusion about the role of these pathogens in the development of *Pneumocystis* pneumonia is difficult to draw. Laakkonen et al. did not find statistical evidence of any association between *Pneumocystis* and *Protostrongylus* spp. in hares [96] or antibodies against arena-, hanta-, or poxvirus in mice and shrews [92]. In cases with polymicrobial diseases, different pathogen species may be involved at the same time [6,20]. Even if not all of them are directly impairing the immune system, interactions and synergisms may facilitate *Pneumocystis* proliferation. Little information is available on non-infectious concomitant factors in pet, farm, zoo, and wild animals, and the effects of environmental factors or habitat changes on the immune system have not yet been studied in relation to the development of *Pneumocystis* pneumonia. A higher susceptibility of juvenile individuals has been suggested [6,11,18,25,26,28,29,35,36,40,45,46,48,50,51,53,55,97,98,99,100,101,103,104,106,107,108,109,110,137], but could not be tested statistically due to the heterogeneity of data.

Genetic information on *Pneumocystis* derived from pet, farm, zoo, and wild animals is limited. Various research groups focused mainly on the heterogeneity of *Pneumocystis* sequences derived from different hosts and concluded that the Genus possessed an outstanding specificity and co-evolution with its host since the fungus phylogenetic clusters generally reflected the genetic relationships between the different mammalian hosts [13,22,38,45,54,60,67,70,71,72,82,84,88,112,113,114,117,118,119,122,124]. This assumption was recently questioned, as a higher genome synteny was found between *P. wakefieldiae* and *P. murina* than between *P. wakefieldiae* and *P. carinii* [43]. Based on the analysis of mtLSU and mtSSU rRNA genes, *P. wakefieldiae* has thus far been placed as an outgroup of the *P. carinii*/*P. murina* clade [74] or a sister species of *P. carinii* [1]. *P. oryctolagi* was more closely related to *P. jirovecii* and *P. macacae* [43], while the rabbit is genetically more closely related to the rodent order.

*P. wakefieldiae* and *P. carinii* DNA were amplified from wild brown rats captured in Thailand. The two strains occurred as single infections, but also co-infections with both were detectable in two samples. Sequencing revealed the presence of a third variant that could neither be assigned to *P. wakefieldiae* nor to *P. carinii* [124]. Besides these two rat-derived *Pneumocystis* species, the forma specialis *P.*sp. *rattus*, *P.* sp. *rattus-secundi*, *P*. sp. *rattus-tertii*, and *P*. sp. *rattus-quarti* were described in wild brown rats. Co-infections with *P. carinii* and these variants could be detected [123]. The presence of genetically distinct *Pneumocystis* variants within one host has furthermore been described in dogs [43,47], cats [72], pigs [20,24], European woodmice [117], and wild Southeast Asian rodents [118,119]. Latinne et al. (2021) [118] determined an association between habitat and the number of variants infecting one host. Synanthropic and generalist rodent species living in heterogenous habitats harbored various *Pneumocystis* lineages, whereas rodent species with a strong preference for forest habitats or dry fields were infected by single *Pneumocystis* variants and did not significantly contribute to the *Pneumocystis* cross-species transmission. Genetic information, especially of *Pneumocystis* species derived from wild animals, is scarce, but *Pneumocystis* cross-species transmission may not be the exception [154,155].

## 5. Conclusions

Although wildlife sampling is complex, as many species are difficult to access, this was the housing category with the most available data. Studies on pet and zoo animals were published to an equal amount, farm animals were at the tail end. Even though the range of published *Pneumocystis* prevalence data was huge, in half of the ICs, a prevalence of ≤ 25% was documented. Prevalence was significantly dependent on the method used, with molecular methods resulting in extraordinarily high positive rates. There was no correlation between prevalence and sample size. Nevertheless, more than half of the studies used sample numbers below or equal to ten, and the utilization of adequate sample sizes is of tremendous importance for the acquisition of reliable and representative research results. Although of enormous significance for distinguishing pure colonization from infection, the level of infection was only indicated in a quarter of the ICs. The higher proportion of high-grade infection levels found in pet and farm animals may be a result of selective breeding for specific traits or better performance causing congenital immunosuppression. In wild animals, the impairment of the immune system could be the result of a reduced habitat, a reduced or suboptimal food supply, or inbreeding. Since high-grade infection levels were associated with a higher *Pneumocystis* prevalence, the expulsion of the fungus may be stronger in severe cases and the fungus may more easily spread to a higher number of new animals. In contrast to loners, wild animals living in groups may be better protected by group/herd immunity. The influence of social and environmental factors on *Pneumocystis* transmission has been studied only rudimentarily. Since our meta-analysis supported a potential correlation, prospective studies considering social structure, habitat, and lifestyle factors are of high importance. Although the publication of the pathoanatomical findings is the exception, the results reported were relatively uniform. Histological findings were described more frequently and confirm the presence of only mild lung lesions which could be related to the strong host dependence of the fungus. The description of the fungal organisms was also very consistent. An underlying immunosuppression was rarely confirmed by laboratory methods but in most of the cases, severe *Pneumocystis* pneumonia was reported. Only some dog and horse breeds have been associated with congenital immunosuppression and in the respective ICs, mainly high-grade infection levels were documented. In contrast, co-infections were, to a large extent, also associated with low-grade infections. Hence, the impact of co-infections, especially of highly immunosuppressive ones, still must be investigated in depth. Individual severe cases of *Pneumocystis* pneumonia were associated with non-infectious diseases. The genetic heterogeneity of individual *Pneumocystis* species has already been shown. However, previous results indicate that heterogeneity might be even larger than assumed and we might only have a vague idea of the actual dimensions of the genetic diversity within the Genus Pneumocystis.

## Figures and Tables

**Figure 1 jof-09-01081-f001:**
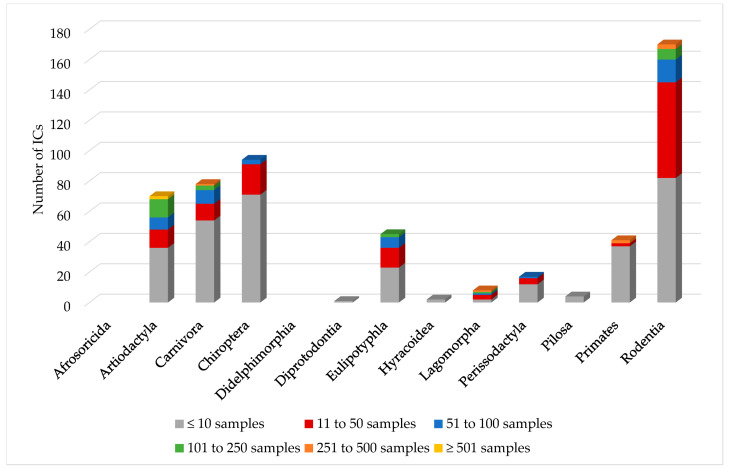
Examined sample numbers of the different mammal orders.

**Figure 2 jof-09-01081-f002:**
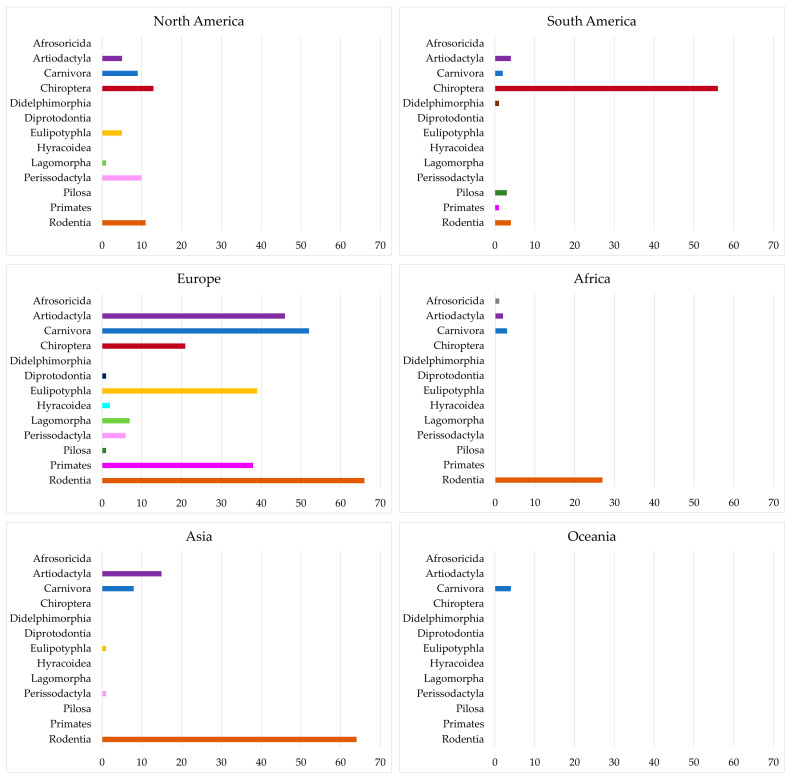
Number of ICs originating from different continents (Mammal orders of Afrosoricida in grey, Artiodactyla in violet, Carnivora in light blue, Chiroptera in red, Didelphimorphia in brown, Diprotodontia in dark blue, Eulipotyphla in yellow, Hyracoidea in turquoise, Lagomorpha in light green, Perissodactyla in rose, Pilosa in dark green, Primates in pink, and Rodentia in orange).

**Figure 3 jof-09-01081-f003:**
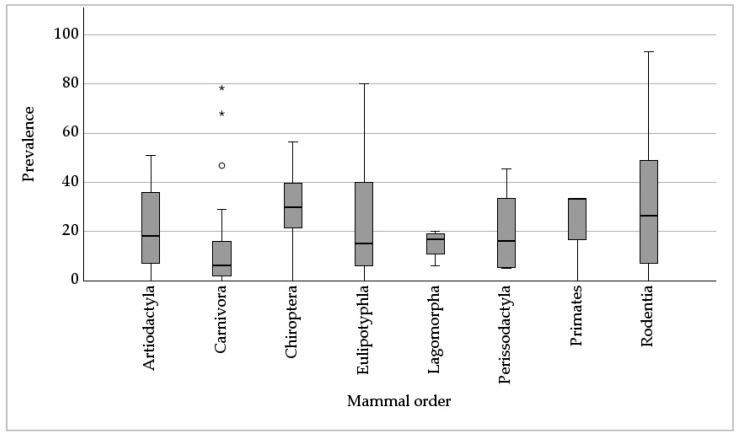
Boxplot of *Pneumocystis* prevalence in eight mammal orders (rings and asterisks represent mild and extreme outliers, ◦ = values between inner and outer fence [1.5-fold interquartile range], * = values beyond outer fence [3-fold interquartile range]).

**Figure 4 jof-09-01081-f004:**
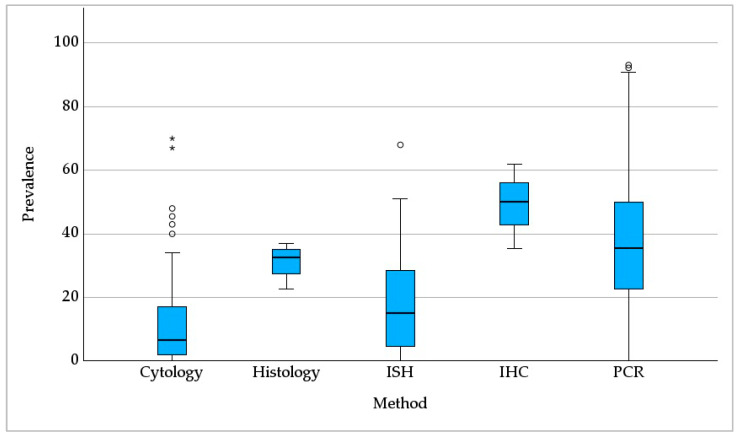
Boxplot of *Pneumocystis* prevalence acquired via different methods (rings and asterisks represent mild and extreme outliers, ◦ = values between inner and outer fence [1.5-fold interquartile range], * = values beyond outer fence [3-fold interquartile range]).

**Figure 5 jof-09-01081-f005:**
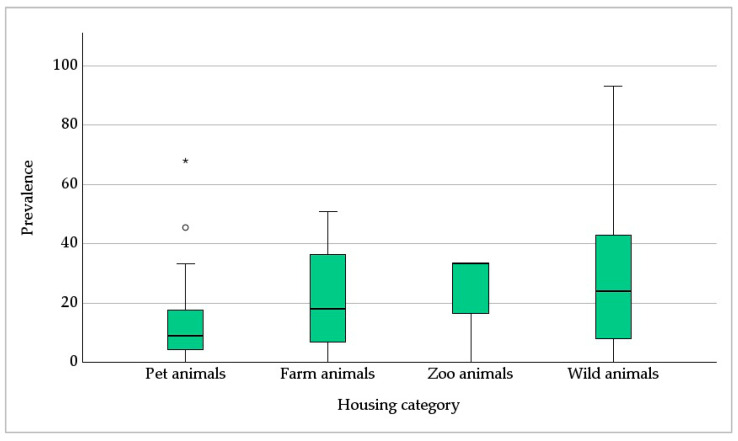
Boxplot of *Pneumocystis* prevalence in different housing categories (rings and asterisks represent mild and extreme outliers, ◦ = values between inner and outer fence [1.5-fold interquartile range], * = values beyond outer fence [3-fold interquartile range]).

**Figure 6 jof-09-01081-f006:**
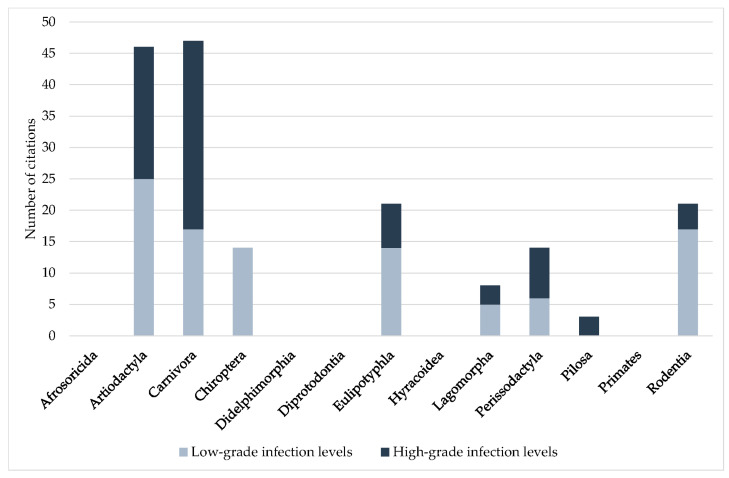
Distribution of low- (light gray) and high-grade (dark gray) *Pneumocystis* infection levels in the 13 investigated mammal orders.

**Figure 7 jof-09-01081-f007:**
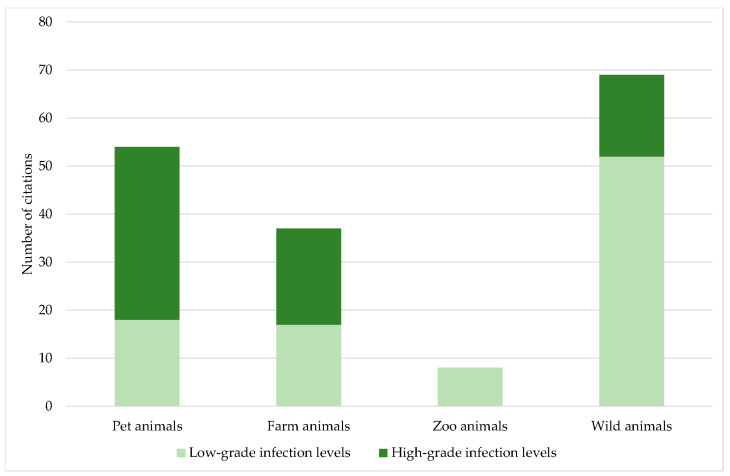
Distribution of low- (light green) and high-grade (dark green) infection levels in pet, farm, zoo, and wild animals.

**Figure 8 jof-09-01081-f008:**
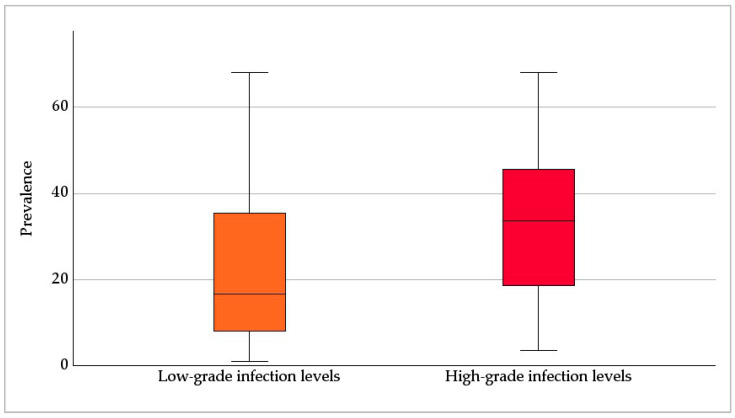
Boxplot of *Pneumocystis* prevalence in relation to low- and high-grade infection levels.

**Figure 9 jof-09-01081-f009:**
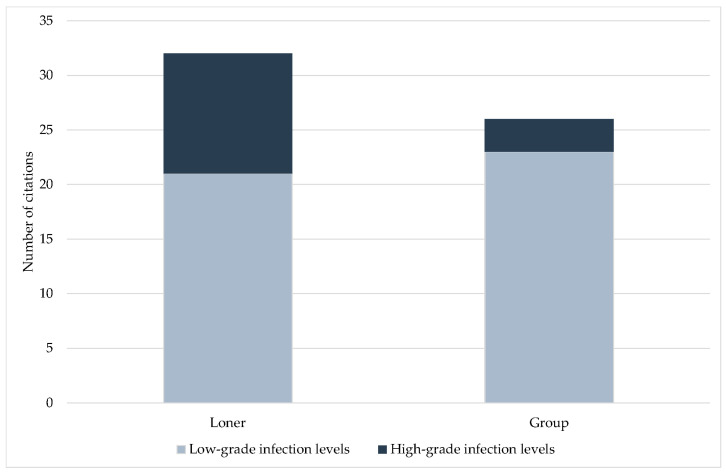
Distribution of low- (light gray) and high-grade (dark gray) infection levels in loners compared to animals living in groups.

**Figure 10 jof-09-01081-f010:**
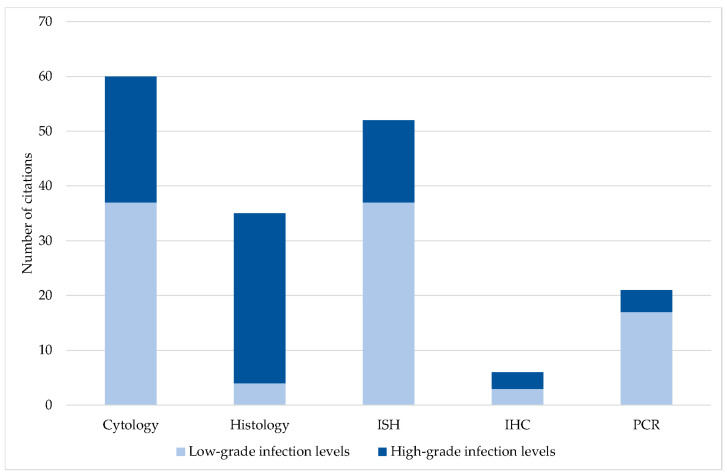
Proportion of low- (light blue) and high-grade (dark blue) infection levels in the different methods used for *Pneumocystis* detection.

**Table 1 jof-09-01081-t001:** Exclusion criteria and final numbers of ICs used for statistical evaluation.

Parameter	ICs Total	Exclusion Criteria	ICs Final
Prevalence	539	Sample sizes ≤ 10 samples (*n* = 324)	187
		No exact sample size indicated (*n* = 7)	
		No number of positive samples indicated (*n* = 1)	
		Selected pre-defined samples (*n* = 1)	
		Studies on exclusively positive samples (*n* = 1)	
		Same samples used as in previous study (*n* = 1)	
		Species not exactly defined (*n* = 5)	
		Serological studies (*n* = 11)	
		Method not defined (*n* = 1)	
Prevalence in wild animals	332	Sample sizes ≤ 10 samples (*n* = 180)	133
		No exact sample size indicated (*n* = 7)	
		No number of positive samples indicated (*n* = 0)	
		Selected pre-defined samples (*n* = 0)	
		Studies on exclusively positive samples (*n* = 0)	
		Same samples used as in previous study (*n* = 0)	
		Species not exactly defined (*n* = 5)	
		Serological studies (*n* = 7)	
Level of infection	539	Only negative results (*n* = 148)	139
		Level of infection not indicated (*n* = 249)	
		No exact sample size indicated (*n* = 0)	
		No number of positive samples indicated (*n* = 0)	
		Same samples used as in previous study (*n* = 0)	
		Species not exactly defined (*n* = 3)	
		Serological studies (*n* = 0)	
		Method not defined (*n* = 0)	
Level of infection in wild animals	332	Only negative results (*n* = 88)	58
		Level of infection not indicated (*n* = 184)	
		No exact sample size indicated (*n* = 0)	
		No number of positive samples indicated (*n* = 0)	
		Same samples used as in previous study (*n* = 0)	
		Species not exactly defined (*n* = 2)	
		Serological studies (*n* = 0)	
		Method not defined (*n* = 0)	
Prevalence x level of infection	187	Only negative results (*n* = 23)	60
		Level of infection not indicated (*n* = 104)	

**Table 2 jof-09-01081-t002:** Numbers of included mammal orders, families, species, and numbers of ICs.

Orders	Families (*n*)	Species (*n*)	ICs (*n*)
Afrosoricida (African shrew-like mammals)	1	(+1) *	1
Artiodactyla (even-toed ungulates)	4	24 (+2) *	72
Carnivora (carnivores)	5	29 (+1) *	79
Chiroptera (bats)	7	44 (+4) *	95
Didelphimorphia (didelphis)	1	1	1
Diprotodontia (marsupials)	1	1	1
Eulipotyphla (insectivores)	4	20 (+1) *	45
Hyracoidea (rock rabbits/dassies)	1	2	2
Lagomorpha (lagomorphs)	1	3	8
Perissodactyla (odd-toed ungulates)	1	3	17
Pilosa (placentals)	1	3	4
Primates (primates)	9	32 (+2) *	41
Rodentia (rodents)	15	96 (+7) *	173
Sum: 13	51	258 (+18) *	539

* (*n*) exact species name not available.

**Table 3 jof-09-01081-t003:** List of mammal orders, families, and species.

Order	Family	Species
Afrosoricida (African shrew-like mammals)	Tenrecidae (tenrecs)	*Microgale* spp. (shrew tenrec) [5]
Artiodactyla (even-toed ungulates)	Bovidae (ruminants)	*Antilope cervicapra* (blackbuck) [6], *Antilope* spp. (antelope) [7], *Bos bonasus* (bison) [6,7], *Bos taurus* (cattle) [6,8,9,10,11], *Bubalus bubalis* (water buffalo) [6], *Capra hircus* (goat) [6,8,11,12,13], *Hippotragus niger* (sable antelope) [14], *Ovis aries* (sheep) [6,7,11], *Rupicapra rupicapra* (chamois) [6]
	Camelidae (camelids)	*Camelus bactrianus* (Bactrian camel) [6], *Camelus dromedarius* (Arabian camel) [6], *Lama glama* (llama) [6,15], *Vicugna pacos* (alpaca) [6]
	Cervidae (deer)	*Capreolus capreolus* (Western roe deer) [6,16], *Cervus alfredi* (Visayan spotted deer) [6], *Cervus elaphus* (red deer) [6], *Cervus nippon* (Sika deer) [6], *Cervus* spp. (deer) [6,9], *Dama dama* (European fallow deer) [11], *Elaphurus davidianus* (Pere David’s deer) [6], *Muntiacus muntjak* (Indian muntjac) [7], *Rangifer tarandus* (reindeer) [6], *Rangifer tarandus groenlandicus* (caribou) [16], *Rusa unicolor* (sambar) [7]
	Suidae (swine)	*Sus scrofa* (wild boar) [6,9,17], *Sus scrofa domesticus* (domestic pig) [6,8,9,10,11,18,19,20,21,22,23,24,25,26,27,28,29,30,31,32,33,34,35,36]
Carnivora (carnivores)	Ailuridae (red pandas)	*Ailurus fulgens* (lesser panda) [37]
	Canidae (canids)	*Canis aureus* (golden jackal) [38], *Canis lupus* (gray wolf) [6], *Canis lupus familiaris* (dog) [6,9,21,39,40,41,42,43,44,45,46,47,48,49,50,51,52,53,54,55,56,57,58,59,60,61,62,63,64,65,66,67,68,69], *Canis lupus lycaon* (Eastern Canadian wolf) [6], *Nyctereutes procyonoides* (raccoon dog) [70], *Vulpes vulpes* (red fox) [6,7,9,16,70,71], *Vulpes zerda* (fennec fox) [37]
	Felidae (felids)	*Felis catus* (cat) [6,9,21,64,69,72,73], *Felis silvestris* (wild cat) [6], *Lynx lynx* (Eurasian lynx) [6], *Panthera leo* (lion) [6], *Panthera pardus* (leopard) [6], *Panthera tigris* (tiger) [6], *Puma concolor* (puma) [6,7]
	Mustelidae (mustelids)	*Aonyx cinereus* (Oriental small-clawed otter) [6], *Galictis vittate* (greater grison) [74], *Lontra canadensis* (Northern American river otter) [6], *Lutra lutra* (Eurasian river otter) [6], *Martes foina* (beach marten) [6,16], *Martes* spp. (marten) [6], *Meles meles* (Eurasian badger) [6,16,71], *Mephitis mephitis* (striped skunk) [6], *Mustela lutreola* (European mink) [6] *Mustela nivalis* (least weasel) [75], *Mustela putorius furo* (ferret) [6,76], *Mustela sibirica* (Siberian weasel) [9], *Neogale vison* (American mink) [77]
	Procyonidae (procyonids)	*Nasua narica* (white-nosed coati) [78], *Procyon lotor* (raccoon) [6]
Chiroptera (bats)	Molossidae (free-tailed bats)	*Eumops glaucinus* (Wagner’s bonneted bat) [79,80], *Molossus currentium* (Thomas’s mastiff bat) [79,80], *Molossus molossus* (Pallas’s mastiff bat) [79,80,81], *Molossus rufus* (black mastiff bat) [79,80], *Nyctinomops laticaudatus* (broad-eared bat) [79,80,81], *Nyctinomops macrotis* (big free-tailed bat) [79,80], *Promops nasutus* (brown mastiff bat) [79,80], *Tadarida brasiliensis* (Brazilian free-tailed bat) [79,80,81,82,83,84]
	Mormoopidae (mustached, ghost-faced, and naked-backed bats)	*Mormoops megalophylla* (ghost-faced bat) [82,83], *Pteronotus davyi* (Davy’s naked-backed bat) [82,83], *Pteronotus parnellii* (Parnell’s mustached bat) [82,83]
	Natalidae (funnel-eared bats)	*Natalus stramineus* (Mexican greater funnel-eared bat) [82,83]
	Pteropodidae (megabats)	*Cynopterus* spp. (short-nosed fruit bat) [6], *Pteropus giganteus* (Indian flying fox) [6], *Pteropus rodricensis* (Rodriguez flying fox) [82], *Pteropus* spp. (flying fox) [6], *Rousettus aegyptiacus* (Egyptian rousette) [82]
	Phyllostomatidae (New World leaf-nosed bats)	*Artibeus fimbriatus* (fringed fruit-eating bat) [79,80,81], *Artibeus hirsutus* (hairy fruit-eating bat) [82,83], *Artibeus lituratus* (great fruit-eating bat) [79,80], *Artibeus* spp. (fruit-eating bat) [79,80], *Carollia perspicillata* (Seba’s short-tailed bat) [79,80,82,83], *Desmodus rotundus* (common vampire bat) [79,80,81], *Diaemus youngii* (white-winged vampire bat) [79,80], *Diphylla ecaudata* (hairy-legged vampire bat) [80,81], *Glossophaga soricina* (Pallas’s long-tongued bat) [74,79,80,82,83,84], *Rhinophylla pumilio* (dwarf little fruit bat) [80], *Sturnira lilium* (little yellow-shouldered bat) [79,80,81]
	Rhinolophidae (horseshoe bats)	*Rhinolophus hipposideros* (lesser horseshoe bat) [6]
	Vespertilionidae (common bats)	*Aeorestes cinereus* (hoary bat) [80], *Eptesicus furinalis* (Argentine brown bat) [79,80], *Eptesicus serotinus* (common serotine) [6,82], *Histiotus velatus* (tropical big-eared brown bat) [79,80], *Hypsugo savii* (Savi’s pipistrelle) [6], *Lasiurus blossevillii* (Western red bat) [79,80], *Myotis californicus* (California myotis) [82,83], *Myotis daubentoni* (Daubenton’s myotis) [82], *Myotis levis* (yellowish myotis) [79,80,81], *Myotis myotis* (greater mouse-eared bat) [82], *Myotis mystacinus* (whiskered bat) [6], *Myotis nigricans* (black myotis) [79,80], *Nyctalus leisleri* (lesser noctule) [82], *Nyctalus noctula* (noctule) [6,82], *Pipistrellus pipistrellus* (common pipistrelle) [82,84], *Plecotus auritus* (brown big-eared bat) [82], *Plecotus austriacus* (gray big-eared bat) [82], *Vespertilio murinus* (particolored bat) [6], *Vespertilio* spp. (common bat) [6]
Didelphimorphia (didelphis)	Didelphidae (opossums)	*Marmosa murina* (murine mouse opossum) [74]
Diprotodontia (marsupials)	Macropodidae (marsupials)	*Osphranter rufus* (red kangaroo) [37]
Eulipotyphla (insectivores)	Erinaceidae (hedgehogs)	*Atelerix albiventris* (Middle-African hedgehog) [6], *Erinaceus europaeus* (Western European hedgehog) [6,16], *Erinaceus roumanicus* (Northern white-breasted hedgehog) [6]
	Soricidae (shrews)	*Blarina brevicauda* (short-tailed shrew) [85], *Crocidura leucodon* (bicolored shrew) [6], *Crocidura suaveolens* (lesser white-toothed shrew) [6,86], *Neomys fodiens* (Eurasian water shrew) [10,86], *Notiosorex crawfordi* (desert shrew) [87], *Sorex alpinus* (Alpine shrew) [86], *Sorex antinorii* (Valais shrew) [71], *Sorex araneus* (European shrew) [5,6,10,86,88,89,90,91,92,93,94], *Sorex caecutiens* (Laxmann’s shrew) [91,93], *Sorex cinereus* (Cinereus shrew) [85], *Sorex fumeus* (smoky shrew) [85], *Sorex isodon* (even-toothed shrew) [5], *Sorex minutissimus* (miniscule shrew) [5], *Sorex minutus* (Eurasian pygmy shrew) [5,6,10,86,93], *Sorex ornatus* (ornate shrew) [87], *Sorex* spp. (shrew) [95]
	Talpidae (moles)	*Talpa europaea* (European mole) [6,88]
	Tupaiidae (tree shrews)	*Tupaia glis* (common tree shrew) [6,37]
Hyracoidea (rock rabbits/dassies)	Procaviidae (hyraxes)	*Dendrohyrax arboreus* (Southern tree hyrax) [37], *Procavia capensis* (Cape rock hyrax) [37]
Lagomorpha (lagomorphs)	Leporidae (rabbits and hares)	*Lepus europaeus* (European brown hare) [6,16,96,97], *Lepus timidus* (mountain hare) [96], *Oryctolagus cuniculus* (rabbit) [6,21,37]
Perissodactyla (odd-toed ungulates)	Equidae (horses)	*Equus asinus* (donkey) [6], *Equus caballus* (horse) [6,25,98,99,100,101,102,103,104,105,106,107,108,109,110], *Equus quagga* (plains zebra) [6]
Pilosa (placentals)	Bradypodidae (sloths)	*Bradypus tridactylus* (pale-throated three-toed sloth) [37,78], *Bradypus variegatus* (brown-throated sloth) [111], *Choloepus didactylus* (Southern two-toed sloth) [78]
Primates (primates)	Aotidae (night monkeys)	*Aotus trivirgatus* (night monkey) [6]
	Atelidae (howler, spider, and woolly monkeys)	*Alouatta fusca* (brown howler monkey) [37], *Ateles belzebuth* (long-haired spider monkey) [37], *Lagothrix lagothricha* (Humboldt’s woolly monkey) [37]
	Callitrichidae (marmosets and tamarins)	*Callimico goeldii* (Goeldi’s marmoset) [112], *Callithrix aurita* (white-eared marmoset) [37], *Callithrix geoffroyi* (Geoffroy’s marmoset) [112], *Callithrix jacchus* (white-tufted ear marmoset) [6,112], *Leontocebus fuscicollis* (brown-headed tamarin) [6], *Leontopithecus rosalia* (golden lion tamarin) [6], *Saguinus fuscicollis* (brown-headed tamarin) [112], *Saguinus imperator* (emperor tamarin) [112], *Saguinus midas* (Midas tamarin) [112], *Saguinus oedipus* (cotton-top tamarin) [6,112]
	Cebidae (capuchin and squirrel monkeys)	*Cebus capucinus* (white-faced sapajou) [6], *Saimiri sciureus* (common squirrel monkey) [6,112]
	Cercopithecidae (Old World monkeys)	*Allenopithecus nigroviridis* (Allen’s swamp monkey) [112], *Cercopithecus hamlyni* (owl-faced monkey) [112], *Cercopithecus nictitans* (white-nosed guenon) [112], *Cercopithecus*/*Miopithecus* spp. (long-tailed monkey) [6], *Colobus guereza* (mantled guereza) [6], *Colobus polykomos* (king colobus) [6], *Macaca fascicularis* (crab-eating macaque) [112,113], *Macaca mulatta* (rhesus monkey) [6,112], *Macaca nemestrina* (pig-tailed macaque) [112], *Macaca sylvanus* (barbary ape) [6], *Theropithecus gelada* (gelada) [6]
	Galagonidae (galagos)	*Galago demidoff* (Demidoff’s Galago) [37], *Galago senegalensis* (Senegal-Galago) [37]
	Hominidae (great apes)	*Pan troglodytes* (chimpanzee) [37], *Pongo* spp. (orang-utan) [6]
	Lemuridae (lemurids)	*Eulemur macaco* (black lemur) [112], *Hapalemur griseus* (bamboo lemur) [112]
	Pitheciidae (titis, saki monkeys, uakaris)	*Pithecia pithecia* (white-faced saki) [6,112]
Rodentia (rodents)	Bathyergidae (African mole-rats)	*Heliophobius argenteocinereus* (silvery mole-rat) [114]
	Castoridae (beavers)	*Castor fiber* (Eurasian beaver) [6]
	Caviidae (cavies)	*Cavia porcellus* (guinea pig) [6,8], *Kerodon rupestris* (rock cavy) [6]
	Chinchillidae (chinchillas)	*Chinchilla lanigera* (long-tailed chinchilla) [6]
	Cricetidae (hamsters and voles)	*Arvicola terrestris* (Eurasian water vole) [94], *Cricetus cricetus* (black-bellied hamster) [6], *Hylaeamys megacephalus* (large-headed rice rat) [78], *Mesocricetus auratus* (golden hamster) [6], *Microtus agrestis* (short-tailed field vole) [10,86,88,94], *Microtus arvalis* (common vole) [10,86], *Microtus californicus* (California vole) [87], *Microtus montebelli* (Japanese grass vole) [115], *Microtus multiplex* (Alpine pine vole) [71], *Microtus subterraneus* (common pine vole) [10,86], *Myodes glareolus* (bank vole) [10,71,86,94], *Myodes rufocanus* (gray red-backed vole) [115], *Myodes smithi* (Smith’s red-backed vole) [115], *Neotoma fuscipes* (dusky-footed woodrat) [87], *Peromyscus boylii* (brush mouse) [87], *Peromyscus californicus* (California mouse) [87], *Peromyscus maniculatus* (North American deer mouse) [87], *Peromyscus* spp. (deer mouse) [116], *Phodopus sungorus* (Dzhungarian hamster) [6], *Reithrodontomys megalotis* (Western harvest mouse) [87]
	Cuniculidae (pacas)	*Agouti paca* (lowland paca) [74]
	Echimyidae (spiny rats)	*Proechimys guyannensis* (Guyenne spiny-rat) [74]
	Gliridae (dormice)	*Eliomys quercinus* (garden dormouse) [88], *Graphiurus murinus* (woodland dormouse) [114], *Graphiurus* spp. (dormouse) [114], *Muscardinus avellanarius* (hazel dormouse) [86]
	Heteromyidae (heteromyids)	*Chaetodipus californicus* (California pocket mouse) [87], *Dipodomys* spp. (kangaroo rat) [116]
	Muridae (murids)	*Acomys ignitus* (fiery spiny mouse) [114], *Acomys muzei* (Muze spiny mouse) [114], *Acomys ngurui* (Nguru spiny mouse) [114], *Acomys wilsoni* (Wislon’s spiny mouse) [114], *Aethomys chrysophilus* (red rock rat) [114], *Aethomys hindei* (Hinde’s rock rat) [114], *Aethomys kaiseri* (Kaiser’s rock rat) [114], *Apodemus agrarius* (Eurasian field mouse) [92], *Apodemus argenteus* (small Japanese field mouse) [115], *Apodemus flavicollis* (yellow-necked field mouse) [10,71,86,92,94], *Apodemus speciosus* (large Japanese field mouse) [115], *Apodemus sylvaticus* (European woodmouse) [10,86,88,94,117], *Apodemus* spp. (field mouse) [10], *Apomys banahao* (Mount Banahao forest mouse) [118], *Bandicota indica* (greater bandicoot rat) [118,119], *Bandicota savilei* (Savile’s bandicoot rat) [118,119], *Berylmys berdmorei* (Berdmore’s Berylmys) [118,119], *Berylmys bowersi* (Bower’s white-toothed rat) [118,119], *Gerbilliscus vicinus* (East African gerbil) [114], *Grammomys surdaster* (African woodland thicket rat) [114], *Hylomyscus arcimontensis* (Arc Mountain wood mouse) [114], *Lemniscomys rosalia* (single-striped grass mouse) [114], *Lemniscomys striatus* (typical striped grass mouse) [114], *Leopoldamys herberti* (long-tailed giant rat) [118,119], *Leopoldamys neilli* (Neill’s Leopoldamys) [118], *Leopoldamys sabanus* (long-tailed giant rat) [118], *Lophuromys kilonzoi* (Kilonzo’s brush furred rat) [114], *Lophuromys makundii* (Makundi’s brush furred rat) [114], *Mastomys natalensis* (African soft-furred rat) [114], *Maxomys surifer* (Indomalayan maxomys) [118,119], *Meriones unguiculatus* (Mongolian gerbil) [6], *Micromys minutus* (European harvest mouse) [92,115], *Mus caroli* (Ryukyu mouse) [118,119], *Mus cervicolor* (fawn-colored mouse) [118,119], *Mus cookie* (Cook’s mouse) [118,119], *Mus minutoides* (Southern African pygmy mouse) [114], *Mus musculus* (house mouse) [6,7,8,10,86,94,115,120,121,122], *Mus* *pahari* (shrew mouse) [118], *Mus saxicola* (spiny mouse) [6], *Mus triton* (gray-bellied mouse) [114], *Myomyscus brockmani* (Brockman’s Myomyscus) [114], *Niviventer fulvescens* (chestnut white-bellied rat) [118,119], *Otomys angoniensis* (Angoni vlei rat) [114], *Praomys delectorum* (delectable soft-furred mouse) [114], *Rattus andamanensis* (Indochinese forest rat) [118,119], *Rattus argentiventer* (rice-field rat) [118,119], *Rattus everetti* (Philippine forest rat) [118], *Rattus exulans* (Polynesian rat) [118,119], *Rattus nitidus* (white-footed Indochinese rat) [118,119], *Rattus norvegicus* (brown rat) [6,8,9,10,94,118,119,123,124], *Rattus rattus* (black rat) [6,8,94], *Rattus sakeratensis* (lesser rice field rat) [118,119], *Rattus tanezumi* (Oriental house rat) [118,119], *Rattus tiomanicus* (Malayan field rat) [118], *Rattus* spp. (rat) [7,8,125,126], *Rhabdomys dilectus* (Mesic four-striped grass rat) [114], *Sundamys muelleri* (Mueller’s giant Sunda rat) [118]
	Myocastoridae (nutrias/river rats)	*Myocastor coypus* (nutria/river rat) [6,71,127]
	Nesomyidae (n/a)	*Saccostomus umbriventer* (pouched mouse) [114], *Steatomys* spp. (fat mouse) [114]
	Octodontidae (degus)	*Octodon degus* (degu) [6]
	Sciuridae (squirrels)	*Callosciurus finlaysonii* (Finlayson’s squirrel) [127], *Funambulus palmarum* (Indian palm squirrel) [37], *Sciurus aestuans* (Guianan squirrel) [74], *Sciurus vulgaris* (Eurasian red squirrel) [6], *Tamias* spp. (chipmunk) [6], *Tamiops swinhoei* (Swinehoe’s striped squirrel) [6]
	Spalacidae (spalacids)	*Cannomys badius* (lesser bamboo rat) [118,119], *Rhizomys pruinosus* (hoary bamboo rat) [118], *Tachyoryctes splendens* (East African mole rat) [114]

## Data Availability

Publicly available data sets were analyzed in this study. This data can be found here: https://doi.org/10.34876/q34b-q773 (accessed on 2 November 2023).

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
