# Peer review of "Meta-Analysis and Systematic Literature Review of the Genus Pneumocystis in Pet, Farm, Zoo, and Wild Mammal Species"

_jof, 2023, doi:10.3390/jof9111081_

Round 1

Reviewer 1 Report

Comments and Suggestions for Authors

In this article, the authors describe and compare the prevalence of Pneumocystis sp in different genders of mammal species. Although the idea of this meta-analysis is original and could be of interest, the methodology has major flaws which make most of the results uninterpretable. 

Major comments :

1. The number of ICs considered in the meta-analysis is not clear and the exclusion cirteria seem not respected : At the beginning 539 ICs but the sample size <10 is an exclusion criteria cited twice l.90 and 92. Yet, in 3.2.2, it is precised that 324/539 ICs have sample size <10. And for all tables/figures, the number of 539 ICs is given... 

To my opinion, sample size<10 should definitely be excluded, more so when authors describe these ICs as case reports. Case reports and epidemic reports should definitely be excluded from this prevalence meta-analysis, otherwise no comparison with other studies can be done...

Moreover, the other exclusion criteria "exact number of positive samples" cited l.92-93 seem not respected either : l.128-130.

2. The prevalence can not be compared with such discrepencies in the detection technique used : serology and IHC evaluate a contact with a fungus while PCR and cytology/histology/ISH evaluate an active infection. Maybe serology evaluations should be excluded from the comparison?

Besides, are the type of method used not always known? L.190-191 : the sum of ICs with the different techniques gives 203 ICs. What about the 336 others? To my opinion, articles that don't give their detection method should be excluded because it has such major impact on the results.

3. Moreover, there are no mention on the sample type used? A biopsy? A nasal sample? there are major difference in detection results depending on the sample type, in addition to the detection technique.

4. The part on low grade/ high grade infections has poor meaning as there are no clear definitions on either of these terms. It seems to be defined either in the original article or by histological interpretation of the authors of this article? How to compare between articles when some are based on eosinophilic infiltrations, other on the fungal load (fungal load on cytology and PCR are not the same at all). There are no mention of symptoms yet it is one colums of the described excel at the beginning.

5. there are no mention in the article on the potential difficulty to detect Pneumocysits in different species considering its co-evaluation. Potential reason why no Pneumocystis have been found in some article?

6. there are no multivariate analysis yet there are a lot of criteria that influence each other, notably the criteria of proximity between animal (for instance in polar regions, animals may be less numerous?). Age of animals is one of the considered Excel column cited in Methods, yet it is never cited afterwards.

Minor comments : 

- Several Pneumocystis word are not in italic.

- l. 109 : why speak of species when some are not precised : genders?

- figure 1 is difficult to understand. why not number of ICs in y-axis?

- figure 8 is difficult to interpret : relationship between prevalence and low/high grade of infection considering that different types of techniques are used both for the grade of infection and for the prevalence.

- l. 295-300 : shoud be revised. It depends on the criteria fixed by the authors.. Cytology don't detect small amont of Pneumocystis thus low-grade infection in PCr are not the same as in cytology. In humans for instance, when a cytology is positive on respiratory sample, the patient is considered to have pneumocystosis.

- part 3.5 : hypogammaglobulinemia is not a risk factor for Pneumocystis pneumonia in humans.

- L. 607-608 : why this hypothesis? usually pneumocystis pneumonia is described as morerelated to the inflammation response than to the fungal load. 

- L. 637-641 : there are no arguments in that sens in the article.

-L. 741 : how is the pneumocystosis defined? No precisions in the article, no reference to symptoms?

Comments on the Quality of English Language

The long sentence from l.32 to l.36 is not clear.

Author Response

Dear Reviewer 1,

please find the answers to your comments in the attached file.

Reviewer 2 Report

Comments and Suggestions for Authors

It is a very interesting and very well-performed review of Pneumocystis spp. infections in animals. The authors deeply analyzed data available on that issue. Meta-analysis revealed what are the factors involved in the transmission of those infections in companion, farm and zoo animals. The involvement of wildlife is also discussed. The paper includes comments on various diagnostic methods and clinical aspects of the treatment. Summarizing: I have read that article several times and I strongly recommend this to be published by Your Institution. I am convinced that this will be a good source of knowledge for scientists and clinicians interested in infectious diseases.

Author Response

Dear Reviewer 2,

please find the answers to your comments in the attached file.

Reviewer 3 Report

Comments and Suggestions for Authors

Really enjoyed this review and an important piece for referring to the many different types of animal PCP reported in the past. Paper was very well written and documented and included the most recent relevant publications on Pneumocystis through 2023. I was able to hear this paper presented at the latest IWOP and the authored I noted had an excellent grasp of the current work presented in this paper and the work was well received by other experts in the field and no questioning of the data presented. 

Author Response

Dear Reviewer 3,

please find the answers to your comments in the attached file.

Reviewer 4 Report

Comments and Suggestions for Authors

This is a very interesting meta-analysis of the literature surrounding Pneumocystis infection/colonization in non-human and non-research mammalian species.  The data are well presented and the authors do a good job explaining the pros and cons of the different ways in which the fungal organisms were identified.  Overall the prevalence of the Pneumocystis in pets, farms animals, zoo animals, and wild animals was quite variable.  However, it was quite interesting to see the range of animals that harbored organisms.  The data raised a few questions including:

1.  Was the age of the animals known or estimated and was there any pattern based on age.  There is some indication that infant animals may be more susceptible to infection than adults

2.  There was some mention of co-infections but it wasn't clear how prevalent these might have been.  Was there any data that would clarify that question?

3.  How prevalent were the extrapulmonary infections?  In humans it is not unheard of, but relatively rare.

Author Response

Dear Reviewer 4,

please find the answers to your comments in the attached file.

Round 2

Reviewer 1 Report

Comments and Suggestions for Authors

In this article, authors analyzed the prevalence of Pneumocystis gender in different mammal species. This is an interesting topic with a detailed methodology. Yet, some bias are not discussed sufficiently.

Major comments:

- Why are the excluded IC still used in parts 3.3 and 3.4 of the results (level of infection and cytology/histology, respectively), notably ICs < 10 individuals? L.276 : "The level of infection  [...] was reported in 26% (139/539) of the ICs". L.339 : "Five per cent (26/539) of the ICs were presented together with a pathomorphological 339 report."

- L.593 : The impact of Pneumocystis host-specificity on the sensitivity of the different detection means is poorly discussed : There is mention of the PCR primers but what about the antigens targeted for IHC? also, the sensitivity of used primers or antigens could be different between species and have an impact on the detected prevalence. 

- L.44 and l. 276 : why is the level of infection relevant to distinguish subclinical infection from severe pneumonia? Are there any reference to support this? The absence of clinical evlaution should be discussed. Maybe histology could be interesting in this context?

- serological analyses and analysis on <10 individuals should figure in the exclusion criteria paragraph, not only in the table 1. and serological analysis are still mentioned in discussion l 589.

 Minor comments : 

l.37 : use of c) for the second time, probably instead of e).

L.47 : maybe add the keywords that were used for the bibliographical research?

Comments on the Quality of English Language

There is no major problem in the English language in my opinion. Just 2 comments:

L.33 : the beginning of the sentence can be difficult to understand "Nevertheless, also lungs of various animal 33 species have been investigated even though the focus was placed on different topics [...].

L. 109 : none of them instead of "not of them"?

Author Response

We thank the reviewer for the thourough review and the valuable comments. Please find here our answers:

In this article, authors analyzed the prevalence of Pneumocystis gender in different mammal species. This is an interesting topic with a detailed methodology. Yet, some bias are not discussed sufficiently.

Major comments:

- Why are the excluded IC still used in parts 3.3 and 3.4 of the results (level of infection and cytology/histology, respectively), notably ICs < 10 individuals? L.276 : "The level of infection  [...] was reported in 26% (139/539) of the ICs". L.339 : "Five per cent (26/539) of the ICs were presented together with a pathomorphological 339 report."

Author: The study is based on a total of 539 citations which were all included in the systematic literature review and the descriptive analysis. The exclusion criteria summarized in Table 1 were considered for statistical analysis and the numbers were adapted. We intended to give an information on the number of ICs in which the infection level or pathomorphological diagnoses were mentioned in relation to the total number of published citations and think this is comprehensible for the reader. The numbers were checked once again, they are correct.

- L.593 : The impact of Pneumocystis host-specificity on the sensitivity of the different detection means is poorly discussed : There is mention of the PCR primers but what about the antigens targeted for IHC? also, the sensitivity of used primers or antigens could be different between species and have an impact on the detected prevalence. 

Author: We completely agree that Pneumocystis species differ strongly genetically and antigenetically. We have expanded this discussion part (Line 692 ff).

- L.44 and l. 276 : why is the level of infection relevant to distinguish subclinical infection from severe pneumonia? Are there any reference to support this? The absence of clinical evaluation should be discussed. Maybe histology could be interesting in this context?

Author: We added references that support that the level of infection is relevant for distinguishing subclinical from severe pneumonia (line 361 ff). The discussion on the absence of clinical evaluation was included in the discussion and associated with the histological findings (line 760 ff).

- serological analyses and analysis on <10 individuals should figure in the exclusion criteria paragraph, not only in the table 1. and serological analysis are still mentioned in discussion l 589.

Author: The exclusion criteria were listed in Materials and Methods (line 123 ff). The sentence related to serological analysis in the discussion was deleted.

 Minor comments : 

l.37 : use of c) for the second time, probably instead of e).

Author. We apologize for this mistake, we have corrected that.

L.47 : maybe add the keywords that were used for the bibliographical research?

Author: The keywords are covered by the following sentence: Pneumocystis, pneumocystosis, and names of zoological orders, families, and species in Latin and different languages were used as search terms (line 72 ff). A list of keywords would be too long to be added.

Comments on the Quality of English Language

There is no major problem in the English language in my opinion. Just 2 comments:

L.33 : the beginning of the sentence can be difficult to understand "Nevertheless, also lungs of various animal 33 species have been investigated even though the focus was placed on different topics [...].

Author: This sentence was modified.

  1. 109 : none of them instead of "not of them"?

Author: This sentence was modified.